# Deep reinforcement learning with relational inductive biases

**Vinicius Zambaldi**,[*] **David Raposo**,[*] **Adam Santoro**,[*] **Victor Bapst**, **Yujia Li**,
**Igor Babuschkin**, **Karl Tuyls**, **David Reichert**, **Timothy Lillicrap**, **Edward Lockhart**,
**Murray Shanahan**, **Victoria Langston**, **Razvan Pascanu**, **Matthew Botvinick**,
**Oriol Vinyals**, **Peter Battaglia**

DeepMind, London, UK
{vzambaldi,draposo,adamsantoro}@google.com

## ABSTRACT

We introduce an approach for augmenting model-free deep reinforcement learning agents with a mechanism for relational reasoning over structured representations, which improves performance, learning efficiency, generalization, and interpretability. Our architecture encodes an image as a *set* of vectors, and applies an iterative message-passing procedure to discover and reason about relevant entities and relations in a scene. In six of seven StarCraft II Learning Environment mini-games, our agent achieved state-of-the-art performance, and surpassed human grandmaster-level on four. In a novel navigation and planning task, our agent's performance and learning efficiency far exceeded non-relational baselines, it was able to generalize to more complex scenes than it had experienced during training. Moreover, when we examined its learned internal representations, they reflected important structure about the problem and the agent's intentions. The main contribution of this work is to introduce techniques for representing and reasoning about states in model-free deep reinforcement learning agents via relational inductive biases. Our experiments show this approach can offer advantages in efficiency, generalization, and interpretability, and can scale up to meet some of the most challenging test environments in modern artificial intelligence.

## 1 INTRODUCTION

Recent deep reinforcement learning (RL) systems have achieved remarkable performance in very challenging problem domains (Mnih et al., 2015; Silver et al., 2016), in large part because of their flexibility in how they learn and exploit the statistical structure underlying observations and reward signals. But the downsides to such flexibility often include low sample efficiency and poor transfer beyond the specifics of the training environment (Zhang et al., 2018; Lake et al., 2017; Kansky et al., 2017). Various structured approaches to RL (e.g. Dzeroski et al. (2001); Driessens & Dzeroski (2004); Diuk et al. (2008); Garnelo et al. (2016)) have attempted to overcome these limitations by explicitly incorporating entity-based and symbolic representations, and specialized building blocks for solving the task at hand. Although these approaches are often highly efficient, they constrain the representations and admissible learning algorithms, they struggle to learn rich representations, and they are therefore confined to relatively simple tasks and data conditions.

To strike favorable tradeoffs between flexibility and efficiency, a number of recent approaches have explored using *relational inductive biases* in deep learning, to reap the benefits of flexible statistical learning and more structured approaches. Methods such as "graph networks" (Scarselli et al., 2009; Li et al., 2015; Battaglia et al., 2018) explicitly represent entities and their relations using using sets and graphs, and perform relational reasoning using learned message-passing (Gilmer et al., 2017) and attention (Vaswani et al., 2017; Hoshen, 2017; Velickovic et al., 2017; Wang et al., 2017) schemes. Because they are implemented using deep neural networks, they can learn transformations from input

---

[*]Equal contribution.

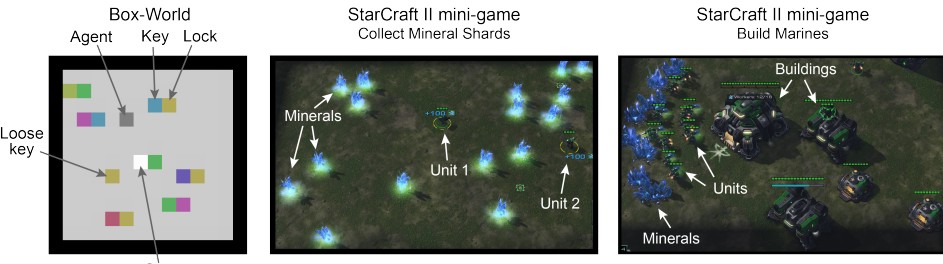

Figure 1: Box-World and StarCraft II tasks demand reasoning about entities and their relations.

observations into task-relevant entities, as well as functions for computing rich interaction among these entities. This provides a powerful capacity for combinatorial generalization, where their learned building blocks can be composed to represent and reason about novel scenarios (Battaglia et al., 2016; Raposo et al., 2017; Dai et al., 2017; Wang et al., 2018; Selsam et al., 2018; Hamrick et al., 2018; Kipf et al., 2018; Sanchez-Gonzalez et al., 2018).

Drawing on several lines of work, we introduce an approach for incorporating relational inductive biases for entity- and relation-centric state representations, and iterated relational reasoning, into a deep RL agent. In contrast with prior work exploring relational inductive biases in deep RL (e.g., Wang et al., 2018), our approach does not rely on *a priori* knowledge of the structure of the problem and is agnostic to the particular relations that need to be considered. To handle raw visual input data, our architecture used a convolutional front-end to compute embeddings of sets of entities, similar to previous work in visual question answering, physical prediction, and video understanding (Santoro et al., 2017; Watters et al., 2017; Wang et al., 2017). To perform relational reasoning, we used a self-attention mechanism (Vaswani et al., 2017; Hoshen, 2017; Velickovic et al., 2017) applied iteratively within each timestep, which can be viewed as learned message-passing (Li et al., 2015; Gilmer et al., 2017). Our deep RL agent is based on an off-policy advantage actor-critic (A2C) method which is very effective across a range of standard RL environments (Espeholt et al., 2018).

Our results show that this relational deep RL agent scale to very challenging tasks, achieving state-of-the-art performance on six out of seven StarCraft II mini-games (Vinyals et al., 2017), surpassing grandmaster level on four mini-games. Additionally, we introduce a novel navigation and planning task, called "Box-World", which stresses the planning and reasoning components of the policy, factoring out other challenges like complex vision or large action spaces. Our agent reaches higher ceiling performance, more efficiently, than non-relational baseline, and is able to generalize to solve problems with more complex solutions than it had been trained on within this task. We also found that the intermediate representations involved in the relational computations were interpretable, and suggest that the agent has rich understanding of the underlying structure of the problem.

## 2 RELATIONAL DEEP RL AGENT ARCHITECTURE

### RL ALGORITHM

We started with a deep RL algorithm based on a distributed advantage actor-critic (A2C) method (Espeholt et al., 2018), which the schematic on the left of Figure 2 summarizes. The agent creates an embedded state representation, $S$ from its input observation, which is then used to compute as output a policy, $\pi$ (the "actor"), and a baseline value, $B$ (the "critic"). The $\pi$ consists of logits over the set of possible actions, from which an action to perform is sampled, and the $B$ is an estimate of the state-value function at the current state. During learning, the $B$ is used to compute the temporal-difference error, which is used both to optimize $\pi$ to generate actions with greater returns than $B$ predicts, and to optimize $B$ to more accurately estimate state values. For further algorithmic details, including the distributed actor/learner framework and off-policy correction methods, please see Espeholt et al. (2018).

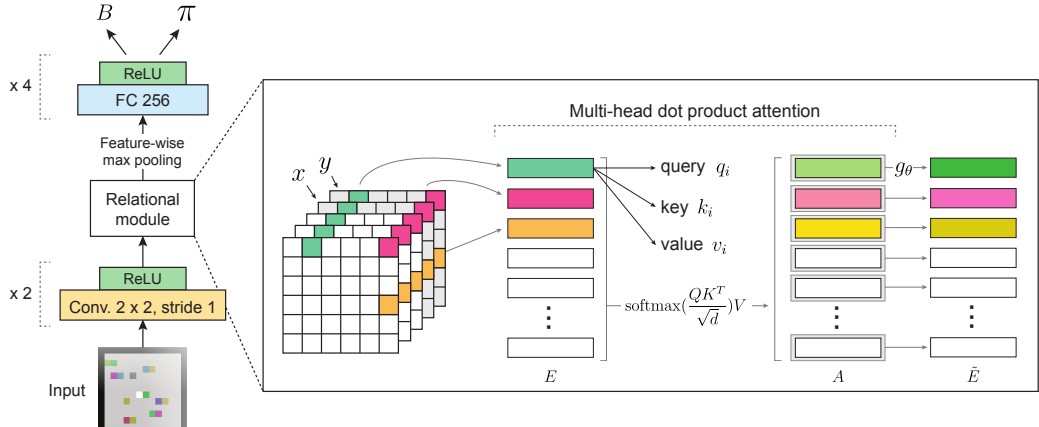

Figure 2: Box-World agent architecture and multi-head dot-product attention. $E$ is a matrix that compiles the entities produced by the visual front-end; $g_\theta$ is a multilayer perceptron applied in parallel to each row of the output of an MHDPA step, $A$, and producing updated entities, $\widetilde{E}$.

INPUT MODULE

Our agent takes as input an image of a scene, transforms it into an embedded state representation, $S$, which is a spatial feature map returned by a convolutional neural network (CNN). The relational module (Figure 2) operated on a set of entities, so our architecture transformed the $m \times n \times f$ feature map into an $N \times f$ (where $N = m \cdot n$) set of "entity vectors", $E$, by reshaping $S$ so that each of $E$'s row, $\mathbf{e}_i$, corresponded to a feature vector, $\mathbf{s}_{x,y}$, at a particular $x, y$ location in the embedded scene. Crucially, this allowed for non-local computation between entities (Wang et al., 2017), unconstrained by their coordinates in the spatial feature map. Because removing the spatial map's coordinate structure could prevent spatial information from being available for downstream computation, we concatenated the spatial $x, y$ coordinates of each $\mathbf{s}_{x,y}$ onto the corresponding $\mathbf{e}_i$ to ensure it was preserved. This feature-to-entity transformation procedure is analogous to how relation networks, visual interaction networks, and the general class of non-local neural networks (Santoro et al., 2017; Watters et al., 2017; Wang et al., 2017) process input images.

RELATIONAL MODULE

To perform one step of relational reasoning, our architecture computed pairwise interactions between each entity and all others (including itself), denoted $\mathbf{p}_{i,j}$, and updated each entity by accumulating information about all of its interactions, $\tilde{\mathbf{e}}_i \leftarrow \{\mathbf{p}_{i,j}\}_{j=1:N}$. This is analogous to how the general class of graph neural networks and their relatives perform entity-wise relational updates (Scarselli et al., 2009; Gilmer et al., 2017). To perform these one-step relational computations, we used self-attention (Vaswani et al., 2017; Hoshen, 2017; Velickovic et al., 2017; Wang et al., 2017), though various other implementations (e.g., those reviewed in Battaglia et al. (2018)) should also be viable.

Our specific self-attention implementation was based on Vaswani et al. (2017)'s multi-head dot-product attention (MHDPA), which we explain in detail to make clear its role as a mechanism of relational reasoning. MHDPA (see Figure 2's expanded box) projects the entities, $E$, into matrices of query, key, and value vectors, $Q$, $K$, and $V$, respectively (which are normalized via "layer normalization" (Ba et al., 2016)). The similarities between query $\mathbf{q}_i$ and all keys, $\mathbf{k}_{j=1:N}$ are computed by a dot-product, which are normalized into attention weights, $\mathbf{w}_i$, via a softmax function, and then used to compute the pairwise interaction terms, $\mathbf{p}_{i,j} = w_{i,j}\mathbf{v}_j$. The accumulation of the interactions for each entity is the sum of these pairwise interactions, $\mathbf{a}_i = \sum_{j=1:N} \mathbf{p}_{i,j}$. This can be efficiently computed using matrix multiplications, $A = \text{softmax}(d^{-\frac{1}{2}} QK^T)V$, where $d$ is the dimensionality of the query and key vectors (i.e. $Q$ and $K$ have shape $N \times d$). Like Vaswani et al. (2017), we also use multiple, independent attention "heads", applied in parallel (which our attention analyses in Results 3.1 suggests the heads may assume distinct relational semantics through training). The updated entities are computed as a function of the accumulated interactions, $\tilde{\mathbf{e}}_i = g_\theta \left( \mathbf{a}_i^{h=1:H} \right)$

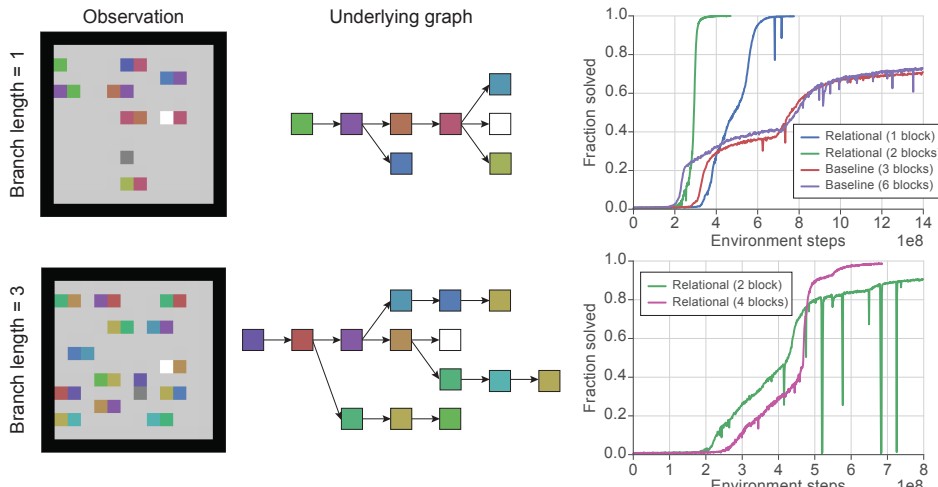

Figure 3: Box-World task: example observations (left), underlying graph structure that determines the proper path to the goal and the distractor branches (middle), and training curves (right).

(where $h$ indexes the head) by concatenating all $\mathbf{a}_i^{h=1:H}$ together, passing them to a multilayer perceptron (MLP), summing the output with $\mathbf{e}_i$ (i.e. a residual connection), and applying a final stage of layer normalization (Ba et al., 2016). We included this non-linear transformation ($g_\theta$) to facilitate the computation of more complex relationships between the entities. This is analogous to what is done in Santoro et al. (2017).

This one-step relational update process, which we term a "block", can be applied iteratively using shared (recurrent) or unshared (deep) parameters, to compute higher order interactions among entities, analogous to message-passing on graphs (Li et al., 2015; Gilmer et al., 2017). Here we refer to the stack of multiple relational blocks as the *relational module*.

### OUTPUT MODULE

The relational module's final output, $\tilde{E}$, is then used to compute $\pi$ and $B$ as follows. The $\tilde{E}$ matrix, with shape $N \times f$, is reduced to an $f$-dimensional vector by max-pooling over the entity dimension. This pooled vector is then passed to a small MLP, which returns an $(c + 1)$-dimensional vector. This vector is split into a $c$-dimensional vector of $\pi$'s logits (where $c$ is the number of discrete actions), and a scalar baseline value estimate, $B$. The $\pi$ logits are normalized using a softmax function, and used as probability parameters for a multinomial distribution, from which an action is randomly sampled.

## 3 EXPERIMENTS AND RESULTS

### 3.1 BOX-WORLD

#### TASK DESCRIPTION

We first introduce a controlled navigation environment, called Box-World[1], which we designed to be perceptually simple, but combinatorially complex, and require abstract relational reasoning and planning. It consists of a $12 \times 12$ pixel room with keys and boxes scattered randomly. The room also contains an agent, represented by a single dark gray pixel, which can move in four directions: *up*, *down*, *left*, *right* (see Figure 1).

Keys are represented by a single colored pixel. The agent can pick up a loose key (i.e., one not adjacent to any other colored pixel) by walking the avatar (i.e. the visual depiction of the agent's location) over it. Boxes are represented by two adjacent colored pixels – the pixel on the right

---

[1]The Box-World environment will be made publicly available online.

represents the box's lock and its color indicates which key can be used to open that lock; the pixel on the left indicates the content of the box which is inaccessible while the box is locked.

To collect the content of a box the agent must first collect the key that opens the box (the one that matches the lock's color), and then walk over the lock, which makes the lock disappear. At this point the content of the box becomes accessible and can be picked up by the agent. Most boxes contain keys that, if made accessible, can be used to open other boxes. One of the boxes contains a gem, represented by a single white pixel. The agent's goal is to acquire the gem by unlocking the box that contains it and picking it up by walking over it. Keys that an agent has in its possession are depicted in the input observation as a pixel in the top-left corner.

In each level there is a unique sequence of boxes that need to be opened in order to reach the gem. Opening one wrong box (a distractor box) consumes the held key and leads to a dead-end, where the gem cannot be reached and the level becomes unsolvable. There are three user-controlled parameters that contribute to the difficulty of the level: 1) the number of boxes in the path to the goal (solution length); 2) the number of distractor branches; 3) the length of the distractor branches. In general, the task is computationally difficult for a few reasons. First, a key can only be used once, so the agent must be able to reason about whether a particular box is along a distractor branch or along the solution path. Second, keys and boxes appear in random locations in the room, demanding a capacity to reason about keys and boxes based on their abstract relations, rather than based on their spatial proximity.

## RL Agents for Box-World

Our agent follows the reinforcement learning setup and architecture described in section (Sec. 2). For a non-relational baseline, we replaced the agent's relational module with a variable number of residual convolution blocks. See the Appendix for further details of the input, relational, and output modules, including hyperparameters and training procedures.

## Results

The training set-up consisted of Box-World levels with solution lengths of at least 1 and up to 4. This ensured that an untrained agent would have a small probability of reaching the goal by chance, at least on some levels.[2] The number of distractor branches was randomly sampled from 0 to 4. Training was split into two variants of the task: one with distractor branches of length 1; another one with distractor branches of length 3 (see Figure 3).

Agents augmented with our relational module achieved close to optimal performance in the two variants of this task, solving more than 98% of the levels. In the task variant with short distractor branches, an agent with a single relational block was able to achieve top performance. In the variant with long distractor branches, a greater number of relational blocks was required, consistent with the conjecture that more blocks allow higher-order relational computations. In contrast, our baseline agents, which can only rely on convolutional and fully-connected layers, performed significantly worse, solving less than 75% of the levels across the two task variants. We observed similar results when we repeated the experiments using two alternative RL algorithms: asynchronous advantage actor-critic (A3C, Mnih et al., 2016) and distributed Q-learning with prioritized experience replay (Horgan et al., 2018). We note that with Q-learning training took significantly longer (see Figure 7 in Appendix).

We also repeated these experiments, but with backward branching in the underlying graph used to generate the level. With backward branching the agent does not need to plan far into the future; when it is in possession of a key, a successful strategy is always to open the matching lock. In contrast, with forward branching scenes, the agent can use a key on the wrong lock (i.e. on a lock along a distractor branch). Thus, forward branching demands more complicated forward planning to determine the correct locks to open, which contrasts with backward branching scenarios where an agent can adopt a more reactive policy, always opting to open the lock that matches the key in possession. Indeed, the baseline agents performed much better in backward (versus forward) branching scenes, which

---

[2]An agent with a random policy solves by chance 2.3% of levels with solution lengths of 1 and 0.0% of levels with solution lengths of 4.

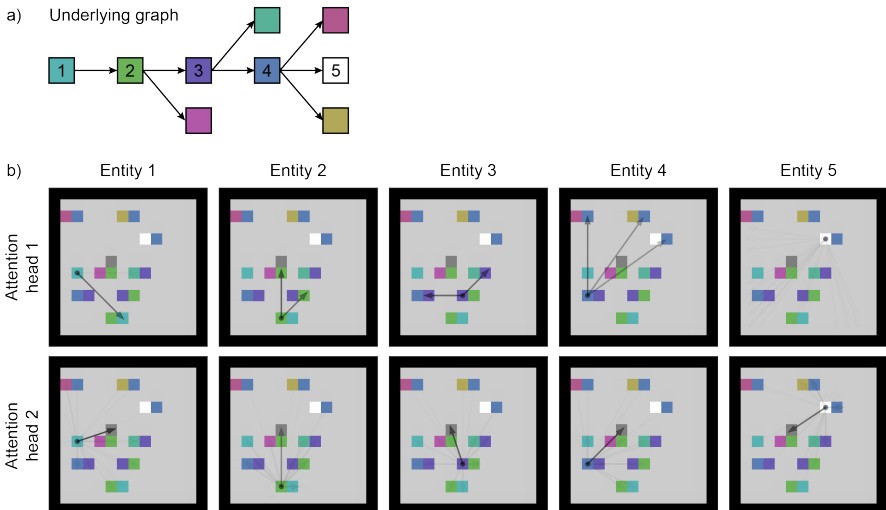

Figure 4: Visualization of attention weights. (a) The underlying graph of one example level; (b) the result of the analysis for that level, using each of the entities along the solution path (1–5) as the *source* of attention. Arrows point to the entities that the *source* is attending to. An arrow's transparency is determined by the corresponding attention weight.

suggests that its weak relational reasoning capacity is to blame for its poor performance in the forward branching condition (see Figure 8 in Appendix).

### VISUALIZATION OF ATTENTION WEIGHTS

We next looked at specific rows of the attention weight matrix, $\mathbf{w}_i$, which corresponded to relevant objects in the scene. Figure 4 shows the result of this analysis when the attending entities (source of the attention) are objects along the solution path. For one of the attention heads, each key attends mostly to the locks that can be unlocked with that key. In other words, the attention weights reflect the options available to the agent once a key is collected. For another attention head, each key attends mostly to the avatar's entity vector. This suggests that object-avatar relations are important, which may help measure of relative position and support navigation.

In the case of RGB pixel inputs, the relationship between keys and locks that can be opened with that key is confounded with the fact that keys and the corresponding locks have the same RGB representation. We therefore repeated the analysis, this time using one-hot representation of the input, where the mapping between keys and the corresponding locks is arbitrary. We found evidence that: 1) keys attend to the locks they can unlock; 2) locks attend to the keys that can be used to unlock them; 3) all the objects attend to the avatar's location; 4) the avatar and gem attend to each other and themselves.

### GENERALIZATION CAPABILITY: TESTING ON WITHHELD ENVIRONMENTS

As we observed, the attention weights captured a link between a key and its corresponding lock, using a shared computation across entities. If the function used to compute the weights (and hence, used to determine that certain keys and locks are related) has learned to represent some general, abstract notion of what it means to "unlock" – e.g., unlocks(key, lock) – then this function should be able to generalize to key-lock combinations that it has never observed during training. Similarly, a capacity to understand "unlocking" shouldn't necessarily be affected by the number of locks that need to be unlocked to reach a solution.

We thus tested the model under two conditions, *without further training*: 1) on levels that required opening a longer sequence of boxes than it had ever observed (6, 8 and 10), and 2) on levels that required using a key-lock combination that was never required for reaching the gem during training, instead only being placed on distractor paths. As shown in Figure 5, in the first condition, the

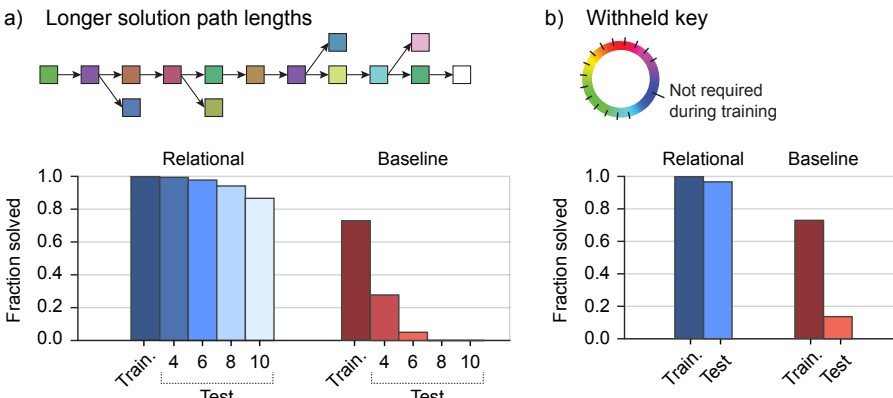

Figure 5: Generalization in Box-World. Zero-shot transfer to levels that required: (a) opening a longer sequence of boxes; (b) using a key-lock combination that was never required during training.

agent with the relational module solved more than 88% of the levels, across all three solution length conditions. In contrast, the performance of the agent trained without the relational module collapsed to 5% when tested on sequences of 6 boxes, and to 0% on sequences of 8 and 10. On levels with new key-lock combinations, the agent augmented with a relational module solved 97% of the new levels. The agent without the relational module performed poorly, reaching only 13%. Together, these results show that the relational module offers a greater capacity for zero-shot transfer to more complex and previously unseen problems. This is likely a consequence of our approach's object- and relation-centric learning, which is less sensitive to the specific conjunctions of objects and relations it has experienced during training.

## 3.2 STARCRAFT II MINI-GAMES

### TASK DESCRIPTION

StarCraft II is a popular video game that presents a difficult challenge for RL agents. It is a multi-agent game where each player controls a large number (hundreds) of units that need to interact and collaborate (see Figure 1). It is partially observable and has a large state and action space, with more than 100 possible actions. The consequences of any single action – in particular, early decisions in the game – are typically only observed many frames later, posing difficulties in temporal credit assignment and exploration.

We trained our agents on the suite of 7 mini-games developed for the StarCraft II Learning Environment (SC2LE, Vinyals et al., 2017). These mini-games were proposed as a set of specific scenarios that are representative of the mechanics of the full game and can be used to test agents in a simpler set up with a better defined reward structure, compared to the full game. The 7 mini-games demand different strategies and skills. Notably, Collect Mineral Shards (CMS), Defeat Roaches (DR) and Defeat Zerglings and Banelings (DZB), require controlling multiple units with precision and in a coordinated manner. To achieve a high score in CMS, the player (or agent) needs to command the two marines independently, defining navigation paths that do not overlap too much in order to efficiently cover most of the ground. In DR, the player must *focus-fire* to succeed, which requires selecting the same attack target, one at a time, for all the marines. In DZB, the enemy units have different abilities, so a good strategy may involve splitting the army into groups with different roles.

### RL AGENTS FOR STARCRAFT II

For StarCraft II (SC2), we start by first constructing a strong control baseline to test against our relational agent. The architecture is similar to that of the Box-World agent, with changes to accommodate specific features of the SC2 environment. We increased the model capacity by using 2 residual convolutional blocks in the input module, each consisting of 3 convolutional layers. We added a 2D-ConvLSTM immediately downstream of the residual blocks, so the subsequent computations were sensitive to the recent history of observations, rather than only the current observation, thus

accounting for partial observability. Including this memory mechanism was important because the actions a StarCraft agent issues to units are carried out over multiple timesteps (e.g., MOVE UNIT 3 TO COORDINATE (1, 5)), and the agent needs a way of not "forgetting" what actions it has issued previously. Otherwise it might keep reissuing the same action to a unit, rather than choosing another action for another unit. Finally, the relational module depicted in section (Sec. 2) was replaced by a stack of residual convolutional blocks in the control agent.

For the agent's output, alongside action $a$ and value $V$, the network produces two sets of action-related arguments: non-spatial arguments ($Args$) and spatial arguments ($Args_{x,y}$). These arguments are used as modifiers of particular actions (see (Vinyals et al., 2017)). $Args$ are produced from the output of the aggregation function, whereas $Args_{x,y}$ result from upsampling the output of the relational module. See the Appendix for further details.

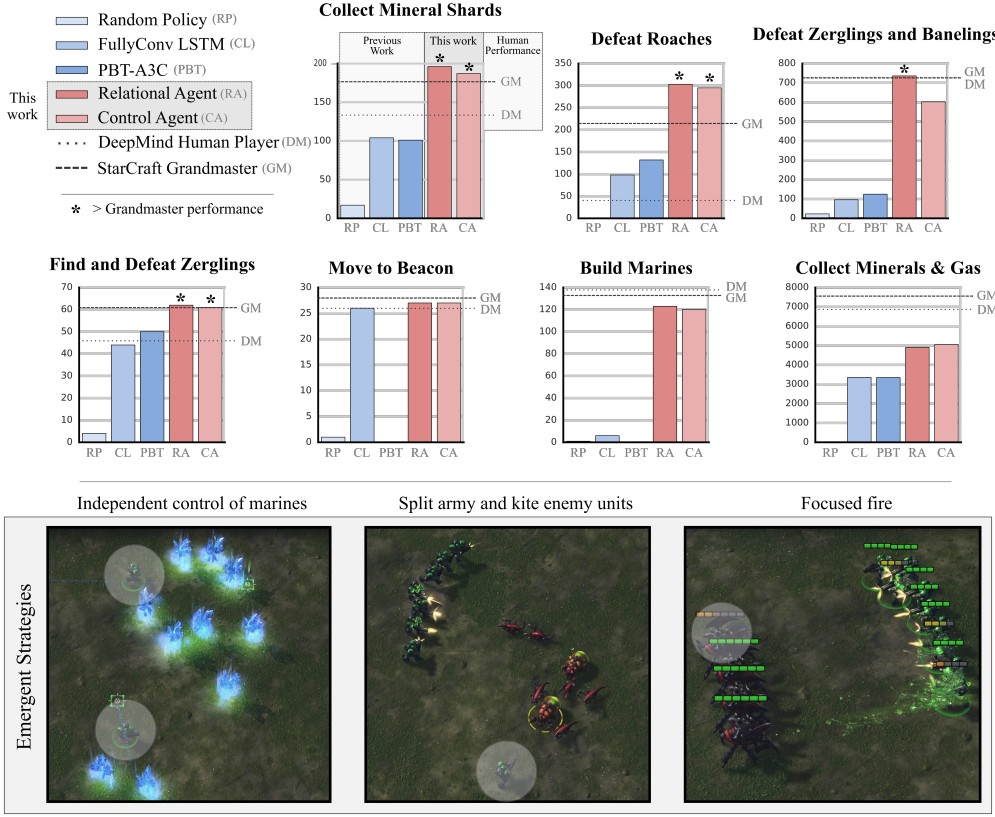

Figure 6: Final performance across StarCraft II mini-games. The relational-agent achieved above grandmaster scores in 4 out of 7 levels. The main difference between our new control agent and the relational agent is most relevant in the Defeat Zerglings and Banelings task. Here, the underlying strategy requires agents to split their army and kite enemy units in order to alleviate the incoming impact (as Banelings incur area damage); a strategy requiring reasoning and fine control over many units. A relational agent with iterative blocks of attention was required for this strategy to emerge. For more details, see Table 1 in the Appendix.

## RESULTS

For these results we used the full action set provided by SC2LE and performance was measured as the mean score over 30 episodes of the best run for each mini-game. Our agent implementations achieved high scores across all the mini-games (Figure 6), consistently outperforming the previous best models. In particular, the agent augmented with a relational module achieved state-of-the-art

results in six mini-games and its performance surpassed that of the human grandmaster in four of them[3].

Head-to-head comparisons between our two implementations show that the agent with the relational component (relational) achieves similar or better results than the one without (control). Concretely, for Collect Mineral Shards, Defeat Roaches and Defeat Zerglings and Banelings, the relational agent improved relative to the control agent by 9, 8 and 134 points, with a standard error of 1.5, 6.4 and 18.2, respectively.

We note that both models improved substantially over the previous best (Vinyals et al., 2017), and in some mini-games, they approach an empirical ceiling performance. This can be attributed to a combination of factors: improved RL algorithm (Espeholt et al., 2018), more robust architecture, better hyperparameter tuning to address issues of credit assignment and exploration, better action selection procedure, and longer training. Next, we focus on differences afforded by relational inductive biases and turn to particular generalization tests to determine the behavioural traits of the control and relational agents.

GENERALIZATION CAPABILITY

As observed in Box-World, a capacity to better understand underlying relational structure – rather than latch onto superficial statistics – may manifest in better generalization to never-before-seen situations. To test generalization in SC2 we took agents trained on Collect Mineral Shards, which involved using two marines to collect randomly scattered minerals and tested them, without further training, on modified levels that allowed the agents to instead control up to *ten* marines. It's worth highlighting the agents have never been exposed to a single observation present in these generalization experiments. Intuitively, if an agent understands that marines are independent units that can be coordinated, yet controlled independently to collect resources, then increasing the number of marines available should only affect the underlying strategy of unit deployment, and should not catastrophically break model performance.

We observed that—at least for medium sized networks—some interesting generalization capabilities emerge, with the best seeds of the relational agent achieving better generalization scores in the test scenario. However, we noticed high variability in these quantitative results, with the effect diminishing when using larger models (which may be more prone to overfitting on the training set). Qualitative analysis of the policies revealed distinct behaviours for the best performing control and relational agents: while the former adopted a "land sweep strategy", controlling many units as a group to cover the space, the latter managed to independently control several units simultaneously, suggesting a finer grained understanding of the game dynamics. Further work is required to draw firm conclusions about the generalization capabilities of a relational agent in more complex domains such as StarCraft II (see Figure 9 in Appendix).

Given the combinatoric richness and multi-agent aspects of the full StarCraft II game, an agent is frequently exposed to situations in which it might not have been trained on. Thus, an improved capability to generalize to new, unseen situations aided by a better understanding of underlying abstract entities and their relations is fundamental.

## 4 CONCLUSION

By introducing structured perception and relational reasoning into deep RL architectures, our agents can learn interpretable representations, and exceed baseline agents in terms of sample complexity, ability to generalize, and overall performance. Behavioral analyses showed that the learned representations allowed for better generalization, which is characteristic of relational representations. Analysis of attention weights showed that the model's internal computations were interpretable, and congruent with the computations we would expect from a model computing task-relevant relations.

One important future direction is to explore ways to scale our approach to larger inputs spaces, without suffering, as this and other approaches do (e.g., Wang et al., 2017; Santoro et al., 2017), from the quadratic complexity that results from considering all input pairs. Possible avenues involve using a distinct attentional mechanisms that scales linearly with the number of inputs (Hoshen, 2017) or

---

[3]For replay videos visit: `http://bit.ly/2kQWMzE`

filtering out unimportant relations (Malinowski et al., 2018). Other future directions include exploring perceiving complex scenes via more structured formats, such as scene graphs (Xu et al., 2017; Chen et al., 2018), which could be powerful additions to our approach's input module. More complex relational modules could be explored, such as richer graph network implementations (Battaglia et al., 2018), learned approaches for inducing compositional programs (Reed & De Freitas, 2015; Parisotto et al., 2017; Allamanis et al., 2017; Devlin et al., 2017) and reasoning about structured data (Neelakantan et al., 2015; Liang et al., 2016), or even explicit logical reasoning over structured internal representations (Evans et al., 2018), drawing inspiration from more symbolic approaches in classic AI. Our approach may also interface well with approaches for hierarchical RL (Vezhnevets et al., 2017), planning (Guez et al., 2018), and structured behavior representation (Huang et al., 2018), so that the structured internal representations and patterns of reasoning can translate into more structured behaviors.

More speculatively, this work blurs the line between model-free agents, and those with a capacity for more abstract planning. An important feature of model-based approaches is making general knowledge of the environment available for decision-making. Here our inductive biases for entity- and relation-centric representations and iterated reasoning reflect key knowledge about the structure of the world. While not a model in the technical sense, it is possible that the agent learns to exploit this relational architectural prior similarly to how an imagination-based agent's forward model operates (Hamrick et al., 2017; Pascanu et al., 2017; Weber et al., 2017). More generally, our work opens new directions for RL via a principled hybrid of flexible statistical learning and more structured approaches.

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

# APPENDIX

## A  BOX-WORLD

### TASK

Each level in Box-world is procedurally generated. We start by generating a random graph (a tree) that defines the correct path to the goal – i.e., the sequence of boxes that need to be opened to reach the gem. This graph also defines multiple distractor branches – boxes that lead to dead-ends. The agent, keys and boxes, including the one containing the gem, are positioned randomly in the room, assuring that there is enough space for the agent to navigate between boxes. There is a total of 20 keys and 20 locks that are randomly sampled to produce the level. An agent receives a reward of $+10$ for collecting the gem, $+1$ for opening a box in the solution path and $-1$ for opening a distractor box. A level terminates immediately after the gem is collected or a distractor box is opened.

The generation process produces a very large number of possible trees, making it extremely unlikely that the agent will face the same level twice. The procedural generation of levels also allows us to create different training-test splits by withholding levels that conform to a particular case during training and presenting them to the agent at test time.

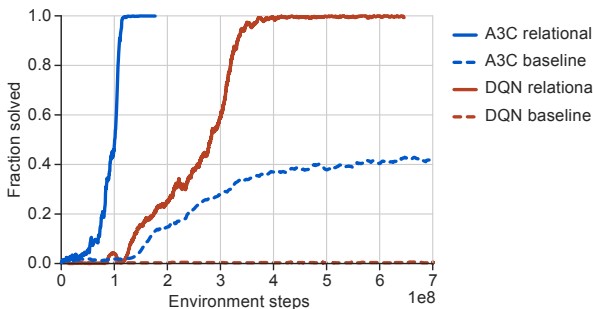

Figure 7: Alternative RL algorithms produced similar results on Box-World. The difference in performance that we observed between *relational* and *baseline* agents using the RL algorithm proposed by Espeholt et al. (2018) was still present when using A3C or distributed DQN. These experiments were done using $10 \times 10$ pixel maps, solution sequences of up to 3 boxes and up to 3 distractor branches.

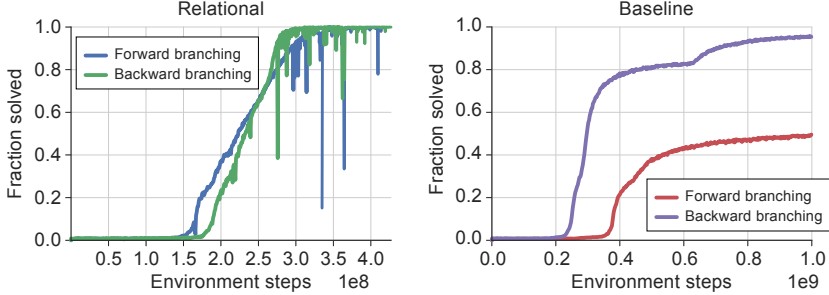

Figure 8: Box-World: forward branching versus backward branching. With backward branching, any given key can only open one box; however, each key type (i.e. color), can appear in multiple boxes. This means that an agent can adopt a more reactive policy without planning beyond which box to open next.

| Agent | Mini-game | | | | | | |
|---|---|---|---|---|---|---|---|
| | 1 | 2 | 3 | 4 | 5 | 6 | 7 |
| DeepMind Human Player (Vinyals et al., 2017) | 26 | 133 | 46 | 41 | 729 | 6880 | 138 |
| StarCraft Grandmaster (Vinyals et al., 2017) | 28 | 177 | 61 | 215 | 727 | 7566 | 133 |
| Random Policy (Vinyals et al., 2017) | 1 | 17 | 4 | 1 | 23 | 12 | < 1 |
| FullyConv LSTM (Vinyals et al., 2017) | 26 | 104 | 44 | 98 | 96 | 3351 | 6 |
| PBT-A3C (Jaderberg et al., 2017) | – | 101 | 50 | 132 | 125 | 3345 | 0 |
| Relational agent | **27** | **196** ↑ | **62** ↑ | **303** ↑ | **736** ↑ | 4906 | **123** |
| Control agent | **27** | 187 ↑ | 61 | 295 ↑ | 602 | **5055** | 120 |

Table 1: Mean scores achieved in the StarCraft II mini-games using full action set. ↑ denotes a score that is higher than a StarCraft Grandmaster. Mini-games: 1–Move To Beacon, 2–Collect Mineral Shards, 3–Find And Defeat Zerglings, 4–Defeat Roaches, 5–Defeat Zerglings And Banelings, 6–Collect Minerals And Gas, 7–Build Marines.

## B  RL TRAINING PROCEDURE

We used distributed A2C agents with off-policy corrections (Espeholt et al., 2018). Each agents consisted of 100 actors generating trajectories of experience, and a single learner, which learns $\pi$ and $B$ using the actors' experiences. The model updates were performed on GPU using mini-batches of 32 trajectories provided by the actors via a queue. The agents used an entropy cost of 0.005, discount ($\gamma$) of 0.99 and unroll length of 40 steps.

Training was done using RMSprop optimiser with momentum of 0, $\epsilon$ of 0.1 and a decay term of 0.99. The learning rate was tuned, taking values between $1e-5$ and $2e-4$.

### AGENT ARCHITECTURE

The input module contained two convolutional layers with 12 and 24 kernels, $2 \times 2$ kernel sizes and a stride of 1, followed by a rectified linear unit (ReLU) activation function. The output was tagged with two extra channels indicating the spatial position ($x$ and $y$) of each cell in the feature map using evenly spaced values between $-1$ and 1. This was passed to the relational module, consisting of relational blocks, with *shared parameters*. Queries, keys and values were produced by 2 to 4 attention heads and had an embedding size ($d$) of 64. The output of this module was aggregated using a feature-wise max pooling function and passed to 4 fully connected layers, each followed by a ReLU. Policy logits ($\pi$, size 4) and baseline function ($B$, size 1) were produced by a linear projection. The policy logits were normalized and used as multinomial distribution from which the action ($a$) was sampled.

### BASELINE AGENT ARCHITECTURE

As a baseline agent we used the same architecture as the relational agent but replaced the relational module with a variable number (3 to 6) of residual-convolutional blocks. Each residual block comprised two convolutional layers, with $3 \times 3$ kernels, stride of 1 and 26 output channels.

## C  STARCRAFT II MINI-GAMES

StarCraft II agents were trained with Adam optimiser for a total of 10 billion steps using batches of 32 trajectories, each unrolled for 80 steps. A linear decay was applied to the optimiser learning rate and entropy loss scaling throughout training (see Table 2 for details). We ran approximately 100 experiments for each mini-game, using the hyperparameter settings indicated in Table 4 combined with 3 seeds.

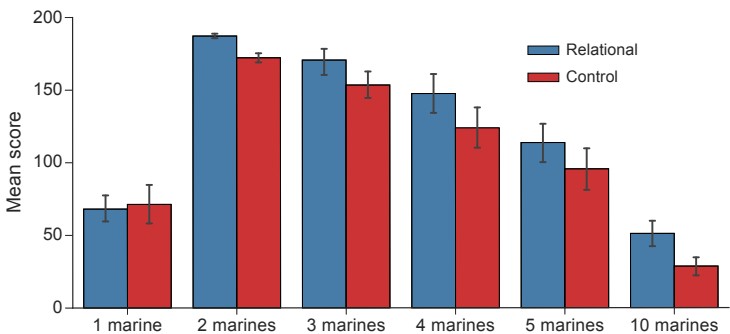

Figure 9: Generalization results on the StarCraft II mini-game Collect Mineral Shards. Agents were trained on levels with 2 marines and tested on levels with 1, 2, 3, 4, 5 or 10 marines. Colored bars indicate mean score over the ten best runs; error bars indicate standard error.

## RELATIONAL AGENT ARCHITECTURE

The StarCraft II (SC2) agent architecture follows closely the one we adopted in Box-World. Here we highlight the changes needed to satisfy SC2 constraints.

**Input-preprocessing**. At each time step agents are presented with 4 sources of information: *minimap*, *screen*, *player*, and *previous-action*. These tensors share the same pre-processing: numerical features are re-scaled with a logarithmic transformation and categorical features are embedded into a continuous 10-dimensional space.

**State encoding**. Spatially encoded inputs (*minimap* and *screen*) are tiled with binary masks denoting whether the previous action constituted a screen- or minimap-related action. These tensors are then fed to independent residual convolutional blocks, each consisting of one convolutional layer ($4 \times 4$ kernels and stride 2) followed by a residual block with 2 convolutional layers ($3 \times 3$ kernels and stride 1), which process and downsample the inputs to $[8 \times 8 \times \#channels_1]$ outputs. These tensors are concatenated along the depth dimension to form a singular spatial input ($inputs_{3D}$, with shape $[8 \times 8 \times \#channels_1 + \#channels_1]$). The remaining inputs (*player* and *previous-action*) are concatenated and passed to a 2-layer MLP (128 units, ReLU, 64 units) to form a singular non-spatial input ($inputs_{2D}$).

**Memory processing**. Next, $inputs_{3D}$ is passed to the Conv2DLSTM along with its previous state to produce a new state and $outputs_{3D}$ (shape $[8 \times 8 \times \#channels_2]$), which represents an aggregated history of input observations.

**Relational processing**. $outputs_{3D}$ is flattened along the first two dimensions (forming a 2D tensor of shape $[64 \times \#channels_2]$ and passed to the stacked MHDPA blocks (see Table 3 for details). Its output tensors (of shape $[64 \times attention\ embedding\ size * number\ of\ attention\ heads]$) follow two separate pathways – *relational-spatial*: reshapes the tensors to their original spatial shape $[8 \times 8]$; *relational-nonspatial*: aggregates through a feature-wise max-pooling operation (combining the 64 embeddings into a flat tensor) and further processes using a 2-layer MLP (512 units per layer, ReLU activations).

**Output processing**. $inputs_{2D}$ and *relational-nonspatial* are concatenated to form a set of *shared features*. Policy logits are produced by feeding *shared features* to a 2-layer MLP (256 units, ReLU, $|actions|$ units) and masking unavailable actions (following Vinyals et al. (2017)). Similarly, baselines values $V$ are generated by feeding *shared features* to a separate 2-layer MLP (256 units, ReLU, 1 unit).

Actions are sampled using computed policy logits and embedded into a 16 dimensional vector. This embedding is used to condition *shared features* and generate logits for non-spatial arguments ($Args$) through independent linear combinations (one for each argument). Finally, spatial arguments ($Args_{x,y}$) are obtained by first deconvolving *relational-spatial* to $[32 \times 32 \times \#channels_3]$ tensors using Conv2DTranspose layers, conditioned by tiling the action embedding along the depth dimension

| Hyperparameter | Value |
|---|---:|
| Conv2DLSTM | |
|     Output channels ($\#channels_2$) | 96 |
|     Kernel shape | (3, 3) |
|     Stride | (1, 1) |
| Conv2DTranspose | |
|     Output channels ($\#channels_3$) | 16 |
|     Kernel shape | (4, 4) |
|     Stride | (2, 2) |
| Discount ($\gamma$) | 0.99 |
| Batch size | 32 |
| Unroll Length | 80 |
| Baseline loss scaling | 0.1 |
| Clip global gradient norm | 100.0 |
| Adam $\beta_1$ | 0.9 |
| Adam $\beta_2$ | 0.999 |
| Adam $\epsilon$ | 1e−8 |

Table 2: Shared fixed hyperparameters across mini-games.

| Setting | Value |
|---|---:|
| MLP layers | 2 |
| Units per MLP layer | 384 |
| MLP activations | ReLU |
| Attention embedding size | 32 |
| Weight sharing | shared MLP across blocks |
| | shared embedding across blocks |

Table 3: Fixed MHDPA settings for StarCraft II mini-games.

and passed by $1 \times 1 \times 1$ convolution layers (one for each spatial argument). Spatial arguments $(x, y)$ are produced by sampling resulting tensors and selecting the corresponding row and column indexes.

CONTROL AGENT ARCHITECTURE

The control agent architecture only differs on the *relational processing* part of the pipeline. Analogous to the relational agent, $outputs_{2D}$ are obtained from Conv2DLSTM layers. These tensors are first passed to a 12-layer deep residual model – comprising 4 blocks of 3 convolutions layers (32 output channels, $4 \times 4$ kernel for the first convolution and $3 \times 3$ for the second and third, and stride 1) interleaved with ReLU activations and skip-connections – as proposed by He et al. (2016), to form the *relational-spatial* outputs. These tensors also follow a separate pathway where they are flattened and passed to a 2-layer MLP (512 units per layer, ReLU activations) to produce what we refer to above as *relational-nonspatial*. The remaining architecture is identical to the relational agent.

| Hyperparameter | Value |
|---|---:|
| Relational module | |
|     Number of heads | $[1, 3]$ |
|     Number of blocks | $[1, 3, 5]$ |
| Entropy loss scaling | $[1e{-}1, 1e{-}2, 1e{-}3]$ |
| Adam learning rate | $[1e{-}4, 1e{-}5]$ |

Table 4: Swept hyperparameters across mini-games.

