# OpenReview forum: "Deep reinforcement learning with relational inductive biases"
_ICLR.cc/2019/Conference_

### Official Review · AnonReviewer3 · 2018-11-02
**A compelling contribution that could benefit from some more quantitative detail**

**Rating:** 7
**Confidence:** 4

**Review:**

The authors present a deep reinforcement learning approach that uses a “self-attention”/“transformer”-style model to incorporate a strong relational inductive bias. Experiments are performed on a synthetic “BoxWorld” environment, which is specifically designed (in a compelling way) to emphasize the need for relational reasoning. The experiments on the BoxWorld environment clearly demonstrate the improvement gained by incorporating a relational inductive bias, including compelling results on generalization. Further experimental results are provided on the StarCraft minigames domain. While the results on StarCraft are more equivocal regarding the importance of the relational module—the authors do set a new state of the art and the results are suggestive of the potential utility of relational inductive biases in more general RL settings.

Overall, this is a well-written and compelling paper. The model is well-described, the BoxWorld results are compelling, and the performance on the StarCraft domain is also quite strong. The paper clearly demonstrates the utility of relational inductive biases in reinforcement learning.

In terms of areas for potential improvement:

1) With regards to framing, a naive reader would probably get the impression that this is the first-ever work to consider a relational inductive bias in deep RL, which is not the case, as the NerveNet paper (Wang et al., 2018) also considers using a graph neural network for deep RL. There are clear differences between this work and NerveNet—most prominently, NerveNet only uses a relational inductive bias for the policy network by assuming that a graph-structured representation is known a priori for the agent. Nonetheless, NerveNet does also incorporate a relational inductive bias for deep RL and shows how this can lead to better generalization. Thus, this paper would be improved by properly positioning itself w.r.t. NerveNet and highlighting how it is different.

2) As with other work using non-local neural networks (or fully-connected GNNs), there is the potential issue of scalability due to the need to consider all input pairs. A discussion of this issue would be very useful, as it is not clear how this approach could scale to domains with very large input spaces.

3) Some details on the StarCraft experiments could be made more rigorous and quantitative. In particular, the following instances could benefit from more experimental details and/or clarifications:

Figure 6: The performance of the control model and relational model seem very close. Any quantitative insight on this performance gap would improve the paper. For instance, is the gap between these two models significantly larger than the average gap between runs over two different random seeds? It would greatly strengthen the paper to clarify that quantitive aspect.

Page 8: ”We observed that—at least for medium sized networks—some interesting generalization capabilities emerge, with the best seeds of the relational agent achieving better generalization scores in the test scenario” — While there is additional info in the appendix, without quantitative framing this statement is hard to appreciate. I would suggest more quantitive detail and rigorous statistical tests, e.g.,  something like “When examining the best 10 out of ??? seeds, the relational model achieved an average performance increase of ???% compared to the control model (p=???, Wilcoxon signed-rank test). However, when examining all seeds ???? was the case.”

Page 8: “while the former adopted a "land sweep strategy", controlling many units as a group to cover the space, the latter managed to independently control several units simultaneously, suggesting a finer grained understanding of the game dynamics.” This is a great insight, and the paper would be greatly strengthened by some quantitive evidence to back it up (if possible). For instance, you could compute the average percentage of agents that are doing the same action at any point in time or within some distance from each other, etc. Adding these kinds of quantitative statistics to back up these qualitative insights would both strengthen the argument, while also making it more explicit how you are coming to these qualitative judgements.

Figure 8 caption: “Colored bars indicate mean score of the ten best seeds” — how bad is the drop to the n-10 non-best seeds? And how many seeds where used in total?

Page 13: “following Table 4 hyperparameter settings and 3 seeds” — if three seeds are used in these experiments, how are 10+?? seeds used for the generalization experiments? The main text implies that the same models for the “Collect Mineral Shards” were re-used, but it appears that many more models with different seeds were trained specifically for the generalization experiment. This should be clarified. Alternatively, it is possible that “seeds” refers to both random seeds and hyperparameter combinations, and it would improve the paper to clarify this. It is possible that I missed something here, but I think it highlights the need for further clarification.

---

> ### Author Response · Authors · 2018-11-16
> **response to reviewer 3**
>
> Thank you for your review! Our goal was precisely to show the utility of relational inductive biases in RL and we are very pleased to know you found the evidence we presented compelling.
>
> Regarding your suggestions:
>
> 1) Thank you for pointing this out. We agree that a mention to NerveNet is justified. We will include a sentence in the text comparing the approaches.
>
> 2) We agree this is a relevant discussion point. As we mentioned in a separate response, using self-attention diminishes the impact of the quadratic complexity compared to other approaches -- e.g Relation Networks (Santoro et al. 2017). This is due to the quadratic computation being reduced to a single matrix multiplication (dot product). Having said this, your point is still a valid one. We are happy to include a discussion point mentioning the scalability challenges and highlight some possible approaches to mitigate this issue.
>
> 3) While we agree that further quantitative detail would benefit the paper, due to the resource intensive nature of StarCraft, we were faced with a harder constraint on the number of hyperparameter and seeds that we could test in each experiment. That being said, we are now running additional tests and computing standard errors to address your points and provide more information about the performance gap between the agents.
>
> Thank you for spotting the incorrect use of the word "seeds" in the caption of Figure 8. To clarify, we ran around 100 combinations of hyperparameters for each mini-game (which included 3 different seeds) as described in page 13. We then used the 10 best runs (not seeds), out of 100, to generate the plot. Regarding the drop in performance after the 10th best run, it follows a linear decay, akin to what we observe for the top 10 runs. We will update the text accordingly to make both points clear.

---

> > ### Comment · AnonReviewer3 · 2018-11-21
> > **Thanks for the response!**
> >
> > Thanks for the thorough response --- I appreciate the additional clarifications being added to the text, and I completely understand that the resource-intensive nature of StarCraft makes some quantitative results difficult to obtain!

---

### Official Review · AnonReviewer1 · 2018-11-03
**Interesting analysis and evaluation of self-attention + relation network in RL; question about novelty**

**Rating:** 7
**Confidence:** 3

**Review:**

This work presents a quantitative and qualitative analysis and evaluation of the self-attention (Vaswani et al., 2017) mechanism combined with relation network (Santoro et al., 2017) in the context of model-free RL. Specifically, they evaluated the proposed relational agent and a control agent on two sets of tasks. The first one “Box-World” is a synthetic environment, which requires the agent to sequential find and use a set of keys in a simple “pixel world”. This simplifies the perceptual aspect and focuses on relational reasoning. The second one is a suite a StarCraft mini-games. The proposed relational agent significantly outperforms the control agent on the “Box-World” tasks and also showed better generalization to unseen tasks. Qualitative analysis of the attention showed some signs of relational reasoning. The result on StarCraft is less significant besides one task “Defeat Zerglings and Banelings". The analysis and evaluation are solid and interesting.

Presentation:
The paper is well written and easy to follow. The main ideas and experiment details are presented clearly (some details in appendix).

One suggestion is that it would help if there can be some quantitive characteristics for each StarCraft task to help the readers understand the amount of relational reasoning required, for example, the total number of objects in the scene, the number of static and moving objects in the scene, etc.

Evaluation:
The evaluation is solid and the qualitative analysis on the “Box-world” tasks is insightful. Two specific comments below:

1. The idea is only compared against a non-relational "control agent”. It would be interesting to compare with other forms of relation networks, for example, the ones used in (Santoro et al, 2017). This could help evaluate the effectiveness of self-attention for capturing interactions.

2. The difference between relational and control agent is quite significant on the synthetic task but less so on the StarCraft tasks, which poses the question of what kind of real-world tasks requires the relational reasoning, and what type of relational reasoning is already captured by a simple non-relational agent.

Question about novelty:

This paper claims it presents “a new approach for representing and reasoning…”. However, the idea of transforming feature map into “entity vectors” and self-attention mechanism are already introduced and the proposed approach is more like a combination of both. That being said, the analysis and evaluation of these ideas in RL are new and interesting.

One minor question: since a level will terminate immediately if a distractor box is opened, does the length of the distractor branches still matter?

Despite the question about novelty, I think the analysis in the paper is solid and interesting. So I support the acceptance of this paper.

Missing references:
In the conclusion section, several related approaches for complex reasoning are discussed. It might be also worth exploring the branch of work (Reed & Freitas, 2015; Neelakantan et al, 2015; Liang et al, 2016) that learns to perform multi-step reasoning by generating compositional programs over structured data like tables and knowledge graph.

Reed, Scott, and Nando De Freitas. "Neural programmer-interpreters." arXiv preprint arXiv:1511.06279 (2015).
Neelakantan, Arvind, Quoc V. Le, and Ilya Sutskever. "Neural programmer: Inducing latent programs with gradient descent." arXiv preprint arXiv:1511.04834 (2015).
Liang, C., Berant, J., Le, Q., Forbus, K. D., & Lao, N. (2016). Neural symbolic machines: Learning semantic parsers on freebase with weak supervision. arXiv preprint arXiv:1611.00020.


Typo:
page 1: "using using sets..."

---

> ### Author Response · Authors · 2018-11-16
> **response to reviewer 1**
>
> Thank you for your thorough review and suggestions, we are grateful you appreciated the work!
>
> To answer your points, one by one:
>
> > Presentation
>
> Thank you for the suggestion. We will add details about each of the StarCraft mini-games in the text to give a better intuition about the task requirements.
>
> > Evaluation
>
> 1) Indeed we ran experiments using the model described in Santoro et al, 2017 as the “relational component” in our agent. We observed that, while the agents were able to learn the task to a certain extent, the training was extremely slow in Box-World and prohibitive in StarCraft-II. We attribute this to the application of a relatively large MLP over each pair of entities (N^2 elements). In fact, this is one of the reasons that attracted us to the multi-head attention to begin with, for its ability to compute pairwise interactions very efficiently -- through a single matrix multiplication (inner product) -- and instead apply an MLP over the resulting N entities (rather than N^2).
>
> 2) We generally agree with your comment. First, it is not obvious the degree to which real-world tasks require explicit relational reasoning. Second, more conventional models, e.g. ConvNets, are capable of a form of relational reasoning, in the sense that they learn the relationships between image patches. Regarding the first point, we have seen recently an increasing number of publications using similar mechanisms to achieve SOTA in a variety of real-world tasks, e.g. visual question answering (Malinowski et al, 2018), face recognition (Xie et al, 2018), translation (Vaswani et al, 2017). This suggests that indeed more explicit ways of comparing/relating different entities helps solving real-world tasks. Regarding the second point, our view is that a capacity to learn relations in a non-local manner (as expressed by Wang et al, 2017) -- i.e. irrespective of how proximal the entities being related are -- will be critical to achieve a satisfying level of generalization in our RL agents. Our results support this hypothesis, but we acknowledge that more work is needed using real-world applications to further establish this idea.
>
> > Novelty
>
> We agree with you that the focus is not on the novelty of these components themselves, but instead on the combination of these for RL, together with careful analyses and evaluation. The sentence you mention might be misleading in that regard and so we propose to change it in the revised version of the paper.
>
> > Length of distractor branches
>
> Yes, the length of the distractor branches still matters. In order for an agent not to take the wrong branch (with perfect confidence) it needs to know the consequences of opening the whole sequence of boxes along that branch before opening the first box in that branch. For that matter, it is irrelevant that the level terminates after the first wrong decision, except for the fact that it reduces the amount of time spent on a level that cannot be solved anymore.
>
> > Missing references
>
> Thank you for the references. These are indeed related to our work and deserve to be mentioned. We will include them.

---

> > ### Comment · AnonReviewer1 · 2018-11-27
> > **Reply to authors' response**
> >
> > Thanks for the response. Most of my concerns are addressed. I think this work is a nice contribution to the community.

---

### Official Review · AnonReviewer2 · 2018-11-06
**Relational Inductive Bias for Deep Reinforcement Learning**

**Rating:** 6
**Confidence:** 4

**Review:**

The goal of this paper is to enhance model-free deep reinforcement techniques with relational knowledge about the environment such that the agents can learn interpretable state representations which subsequently improves sample complexity and generalization ability of the approach. The relational knowledge works as an inductive bias for the reinforcement learning algorithm and provides better understanding of complex environment to the agents.
To achieve this, the authors focus on distributed advantage actor-critic algorithm and propose a shared relational network architecture for parameterizing the actor and critic network. The relational network contains a self-attention mechanism inspired from recent work in that area. Using these new modules, the authors conduct evaluation experiments on  two different environment - synthetic Box World and real-world StarCraft-II minigames where they analyze the performance against non-relational counterparts, visualize the attention weights for interpretability and test on out-of-training tasks for generalizability.

Overall, the paper is well written and provide good explanation of proposed method. The experimental evaluation adequately demonstrates superior performance in terms of task solvability (strong result) and generalizability (to some extent). The idea of introducing relational knowledge into deep reinforcement learning algorithm is novel and timely considering the usefulness of relational representations. However, there are several shortcomings that makes this paper weak:

1.) While it is true that relational representations help to achieve more generalizable approach and some interpretability to learning mechanism, however comparing it to model-based approaches seems a stretch. While the authors themselves present this speculatively in conclusion, they do mention it in abstract and try to relate to model-based approaches.
2.) The relational representation network using pairwise interaction itself is not novel and has been studied extensively. Similarly the self-attention mechanism used in this paper is already available.
3. ) Further, the author chose a specific A2C algorithm to add their relational module. But how about other model-free algorithms? Is this network generalizable to any such algorithm? If yes, will they see similar boost in performance? A comparison/study on using this as general module for various model-free algorithms would make this work strong.
4.) I have some concerns on generalizability claims fro Box World tasks. Currently, the tasks shown are either on levels that require a longer path of boxes than observed or using a key lock combination never used before. But this appears to be a very limited setting. What happens if one just changes the box with a gem between train and test? What happens if the colors of boxes are permuted while keeping the box as it is. I believe the input are parts of scene so how does change in configuration of the scene affect the model's performance?
5.) What is the role of extra MLP g_theta after obtaining A?

Overall it is very important that the authors present some more analysis on use of relational module to generalize across different algorithms or explain the limitations with it. Further it is not clear what are the contributions of the paper other than parameterizing the actor-critic networks with an already known relational and attention module.

---

> ### Author Response · Authors · 2018-11-13
> **response to reviewer 2**
>
> Thank you for your review! We appreciate your suggestions to improve the submission.
>
> To answer each of your points:
>
> 1) We agree that a hard comparison between our approach and model-based planning cannot be made. Our attempt was to bring this as a point of discussion rather than making a strong claim about their parallels. We are happy to revise the text where this is mentioned in the direction of toning down the comparison and avoid confusion.
>
> 2) We tried to be careful throughout the paper not to suggest that the novelty of this work lies on these two components: pairwise interactions and self-attention. Instead, and as mentioned by Reviewer 1, we argue that the combination of learnable representations of entities and self-attention in an RL setting is a significant innovation that has not been attempted before. This was a non-trivial effort, especially when applied to complex RL tasks such as StarCraft-II. Perhaps most importantly, however, it was not clear before that pairwise interactions themselves could allow for improved generalization.
>
> We believe this work is a small but important step that moves us towards addressing some of the criticism deep RL has received (namely, an inability to flexibly generalize 'out-of-distribution') by focusing on entity and relation-centric representations, as used in more symbolic approaches.
>
> 3) Thank you for the suggestion. We agree that showing that the results extend to other model-free algorithms would make the paper stronger. We tested an asynchronous advantage actor-critic (A3C) agent early on and the results were similar, but we will re-run these experiments now, alongside an off-policy value-based RL algorithm (DQN), to get exact numbers.
>
> 4) We appreciate your concerns here. We would like to clarify that indeed the Box-World levels have the features that you propose. Every level is randomly generated in almost every aspect, assuring that: (1) the box containing the gem changes in every level; (2) the colors of the boxes are randomly shuffled in every level; (3) the spatial position of each box is randomly chosen in every level. This random generation of levels makes the problem very hard. In fact the number of possible combinations is so large that the agents we trained on this task never encounter the same level twice. An agent that solves the training levels to 100%, like the relational agent that we proposed, is capable of solving previously unseen levels without making a single mistake.
>
> 5) We found that it was useful to include a shared non-linear transformation over the elements that resulted from the attention mechanism, itself only comprising a weighted sum of elements produced by a single linear transformation. Informally speaking, while the attention produces mixtures of entities, the extra non-linearity (g_theta MLP) gives the model the capacity to compute more complex relationships between the entities. This is analogous to what is done in Relation Networks, by Santoro et al. 2017, described as having the role of “infer[ing] the ways in which two objects are related”. We are happy to include a sentence in the text to provide this intuition.

---

> > ### Comment · AnonReviewer2 · 2018-11-27
> > **Thanks for the response**
> >
> > I believe the authors have addressed most of my comments and the revision has certainly improved the quality of the paper. I still think the overall contribution of the paper is very limited however I agree with the authors that it is indeed an important step towards generalizing RL approaches. In that light, I have adjusted my score and support this paper for acceptance.

---

### Author Response · Authors · 2018-11-26
**paper revision**

We have now submitted a revised version of the paper addressing the criticisms and suggestions from all 3 reviewers. We have also included the results of a new set of experiments using the relational module in combination with different RL algorithms (A3C and distributed DQN), which more clearly demonstrate its general applicability. These results are mentioned in the main text and summarized in Figure 7 in Appendix.

---

### Meta-Review · Area_Chair1 · 2018-12-13
**A significant study of relational inductive biases in DRL**

**Confidence:** 4
**Recommendation:** Accept (Poster)

**Metareview:**

The paper presents a family of models for relational reasoning over structured representations. The experiments show good results in learning efficiency and generalization, in Box-World (grid world) and StarCraft 2 mini-games, trained through reinforcement (IMPALA/off-policy A2C).

The final version would benefit from more qualitative and/or quantitative details in the experimental section, as noted by all reviewers.

The reviewers all agreed that this is worthy of publication at ICLR 2019. E.g. "The paper clearly demonstrates the utility of relational inductive biases in reinforcement learning." (R3)